# The Effectiveness of Various Types of Psychological Correction of Anxiety in Primary School

**DOI:** 10.3390/bs10010020

**Published:** 2019-12-30

**Authors:** Olga P. Bartosh, Tatiana P. Bartosh

**Affiliations:** Scientific Research Center “Arktika”, Far eastern Branch of the Russian Academy of Sciences, Karl Marx Str. 24, Magadan 685000, Russia

**Keywords:** younger schoolchildren, anxiety, therapeutic efforts, sociopsychological training, biological feedback, art therapy

## Abstract

At various stages of the development and education of children, there are psychoemotional difficulties that create prerequisites for impairment of the development of the child’s personality. The timely detection of difficulties in schoolchildren and therapeutic efforts are important for the formation of a psychologically healthy personality. The study of the effectiveness of various remedial techniques for childhood anxiety has therefore become theoretically and practically significant. The purpose of our study is to determine the effectiveness of various types of such therapeutic efforts: social and psychological training, a method of biological feedback, and Sandplay for the indicators of childhood anxiety among younger schoolchildren. The study was conducted in the school of Magadan, northeast of Russia (9–10-year-old students, n = 43). We used a standardized method of Multidimensional Assessment of Child Anxiety which included 10 scales. The following therapeutic efforts were used: sociopsychological training (SPT), biofeedback method (BFB), individual and group Sandplay. Students of group I (n = 12) participated only in the SPT. Students of group II (n = 11) participated in the SPT and underwent a course of training in self-regulation using the BFB method. In therapy work with the students of group III (n = 20), the SPT, BFB, individual and group Sandplay were used. In group I, after the therapy sessions, a significant decrease in anxiety was observed in 3 of 10 scales (2, 6, 7; *p* < 0.05). In group II, it was seen in 5 scales (1, 3, 6, 7, 8; *p* < 0.05). In group III, significant improvements took place in 7 scales (1, 2, 5, 6, 7, 9, 10; *p* < 0.01–*p* < 0.05). The present study has shown the different efficacy of applying the remedial techniques separately and in combination. The use of the therapy methods, in the complex, enhances the impact on the types of child anxiety.

## 1. Introduction

At different stages of the development and education of children, there can be faced various psychoemotional difficulties which create prerequisites for impairment of the child’s personality development. It is known that 75% of the causes for all human diseases appear in childhood [1]. The increase in the number of nonadapted children, especially in primary school, leads to difficulty in solving social and educational tasks. High school anxiety has been the most common marker of school disadaptation. According to many researchers, up to 30–40% of schoolchildren experience discomfort in their school life. These are children with increased school anxiety who are at risk of disadaptation [2]. The stability of the increased anxiety in younger students is often associated with fears and experiences that arise in the situation of schooling. Anxiety is known to be a predictor of depression [3]. It is also of importance that a somatic disease which began during the period of childhood and personality formation can contribute to the development of depression and suicidal manifestations [4,5], psychosomatic and anxiety neurotic disorders [6,7] which are considered the most common disorders all over the world among children and adolescents [8,9]. Of course, increased anxiety influences schoolchildren’s voluntary attention, the amount of random access memory, the speed of perception, mental performance, and academic performance [10].

It is believed that the implementation of school preventive programs in childhood rather than in adolescence provides the best results in the field of mental health [11]. For example, a successful treatment for anxiety disorder in childhood will protect against subsequent suicidality and provide long-term benefits [12].

There are several ways to eliminate the causes of anxiety formed in conditions of socialization under various stresses that influence a person. The key to successful work with an anxious child is an increase in self-esteem, training in self-control and relieving muscle tension [13,14,15].

Sociopsychological training is a multifunctional effective method for developing the child’s psychological characteristics, harmonizing the psychological state, acquiring communication skills, and stimulating the development of self-discovery and self-knowledge. Being important in regulation of behavior and ensuring the best adaptation to life, self-esteem is closely connected with the process of self-consciousness of the individual [16].

Many authors testify that the success of the adaptation of younger schoolchildren and their psychological comfort depends on the ability to manage their mental processes, transforming them into arbitrary, regulated processes that enable them to perform educational tasks more effectively [2]. Psychophysiologically oriented correction and the development of higher mental functions by the method of biological feedback successfully contribute to the therapeutic efforts. According to some researchers [13], conducting health-improving lessons in schools using gaming computer biocontrol makes it possible to facilitate the learning process by the child with the help of acquiring the necessary practical skills of self-regulation in an emotionally tense situation and while performing work on their own.

Among psychologists, Art Therapy is becoming increasingly recognized as a method of therapeutic efforts with children in school. Using Sandplay Therapy helps to relax physically and mentally, reduces impulsivity, excessive physical activity, anxiety, aggression, and children’s fears. Sandplay Therapy allows the child in reality, but on the territory of the psychological sandbox, to work out intrapersonal conflicts, see models of new relationships, increase self-esteem and self-confidence [15,17,18].

The timely detection of difficulties in schoolchildren and the appropriate therapeutic efforts are important for the formation of a psychologically healthy personality. The study of the effectiveness of various remedial techniques for childhood anxiety has therefore become theoretically and practically significant.

The aim of our study is to determine the results of different types of therapeutic efforts such as social and psychological training, method of biological feedback, and art therapy for indicators of childhood anxiety among younger schoolchildren.

## 2. Materials and Methods

### 2.1. Research Design

The study was conducted in the secondary school of Magadan, northeast of Russia (age 9–10 years, n = 43, among them, 25 boys and 18 girls). Diagnostics and therapeutic efforts were carried out before afternoon, in accordance with the principles of the Helsinki Declaration. The study protocol was approved by the Ethical Committee for Biomedical Research at the North-Eastern Scientific Center of the Far-Eastern Branch of the Russian Academy of Sciences (Protocol No.3 of 4 December 2013). The parents of the students were informed about the objectives of the study, and in accordance with the established procedure, they gave informed voluntary consent.

### 2.2. Instruments

To measure the anxiety of younger schoolchildren, we used the MACA method of Multidimensional Assessment of Child Anxiety that includes 10 parameters (“anxieties”) to give a differentiated assessment of anxiety of a person [19]: 1 is general anxiety; 2 is anxiety in relationships with peers; 3 is anxiety in connection with the assessment of others; 4 is anxiety in relationships with teachers; 5 is anxiety in relationships with parents; 6 is anxiety associated with success in learning; 7 is anxiety arising in situations of self-expression; 8 is anxiety in situations of testing knowledge; 9 is decrease in mental activity associated with anxiety; and, finally, 10 is increase in autonomic reactivity associated with anxiety. Total score is from 0 to 10 points. If the parameter is 1–2 points, the quality is considered to be poorly expressed, 4–5 points is for clearly expressed quality, 7–10 points means anxiety is strongly expressed. Also, a total indicator that is an Integral Indicator of Anxiety (IIA) is ascertained.

Based on the results obtained on the 10 scales of MACA, information is received about the structural features of the student’s anxiety in 4 main areas of psychological analysis related to the assessment of: the level of anxiety that is directly related to the personal characteristics of the child (scales 1, 3 and 7); the peculiarities of the psychophysiological and psychoautonomic anxious response of the child in stress situations (scales 9 and 10); the role of the characteristics of the child’s social contacts in the development of anxiety reactions and states (with age mates, teachers, and parents) (respectively, scales 2, 4, and 5); the role of situations related to schooling (scales 6 and 8) in the development of anxiety reactions and conditions of the child.

The following therapeutic efforts were used: sociopsychological training (SPT), biofeedback method (BFB), individual and group art therapy (Sandplay Therapy).

The main methods of SPT are as follows: group discussion, brainstorming, role-playing game, psychogymnastic exercises. Psychological mobile games on contact and friendly attitude to each other, rallying the class, relieving emotional and muscular tension, and developing volitional regulation were proposed. The sequence of exercises involved alternating activities, changing the psychophysical state of a child: from moving to calm, from intellectual play to relaxation techniques. Sociopsychological training was carried out once a week for 2 months.

In our study, the “BOS-Pulse” gaming computer simulator developed at the Institute of Molecular Biology and Biophysics of the Siberian Branch of the Russian Academy of Medical Sciences under the guidance of Academician M.B. Stark was used [20]. The training units of “Vira” and “Rally” were also used. The task of the game training was to teach the child new ways of responding to stressful and conflict situations, mastering the skills of arbitrary regulation of physiological functions under the conditions of psychoemotional stress. The recording of the heart rate (HR) was performed from the nail phalanx of the finger and converted it into feedback signals perceived by the child in the form of a sound or visual row. In order to win the competition, the player must reduce the heart rate by learning to manage their own self-regulation mechanisms in combination with a high degree of control of consciousness. Along with the method of biological feedback, the Progressive muscle relaxation according to Jacobson was applied using breathing techniques and images.

Sandplay therapy involved playing with sand and miniature figures. The child was encouraged to recreate various aspects of the problem in the sand using symbolic objects that could be manipulated. Playing the situation in the psychological sandbox, the child had the opportunity to look at it from the side, to relate the game to real life, to comprehend what was happening, to find constructive solutions to the problem [17,18].

Students of group I (n = 12 children) participated only in the SPT training. Each student participated in 10 training sessions. The main position of the participants was the placement in a circle. Students of group II (n = 11 people) participated in the SPT (10 classes) and completed a course of self-regulation training using the BFB method. The course of the correction classes by the method of BFB consisted of 8–12 sessions for 20–30 min, two times a week. In remedial work with students of group III (n = 20 people), the SPT (10 lessons), BFB (8–12 sessions), individual therapy (3–4 meetings), and group sand therapy were used. The duration of individual sand therapy sessions is one academic hour, followed by group work (4 people each) in the psychological sandbox.

At the final stage, control diagnostics of the level of school anxiety were carried out using the method of MACA in all the students, which made it possible to evaluate the effectiveness of the performed therapeutic efforts.

### 2.3. Statistical Analysis

The obtained data was statistically processed using the software package Statistica 10.0. The results of nonparametric processing methods are presented in the form of a median (Me) and an interquartile range in the form of 25 and 75 percentiles Me (C_25_; C_75_). To test the statistical hypothesis of difference of values, the Wilcoxon criteria for two dependent samples were used. The method of rank correlation by Spearman was used to study the correlations between the studied indices.

## 3. Results

Table 1 presents the indicators of child anxiety in the 3 groups. Considering the data on the total anxiety index (IIA), it can be seen in all the groups that the Integral Indicator of Anxiety 14–21 points decreased (*p* < 0.05–*p* < 0.01).

For students who participated only in psychological training (group I), after completing the course of the training sessions, there were significant 1.5–2 points reductions in the median values, by the 3 MACA scales of 10 (scales 2, 6, 7). Thus, these schoolchildren reduced anxiety in relationships with age mates, improved social contacts, and gained more self-confidence. Anxiety associated with success in learning and in situations of self-expression also proved to decrease.

In group II, after the course of the therapeutic efforts, there were significant 1–4 points reductions in the median anxiety indices by the 5 scales of MACA (1, 3, 6, 7, 8). Moreover, the effect of such therapeutic efforts on the anxiety directly related to the personal characteristics of the child is clearly traced (scales 1, 3, and 7 for general anxiety, anxiety due to the assessment of others, and anxiety arising in situations of self-expression, respectively), and situations related to school education in the development of anxiety reactions and states of the child (scales 6 and 8 for anxiety associated with learning success and anxiety arising in situations of knowledge testing).

Note that, in group II, scale 3, the anxiety suggesting the connection with the assessment of others, has the highest median value of 6.0 (2.5; 7.0) (Table 1). High scores on this scale indicate that the majority of children at this age are experiencing emotional discomfort, tension about the ratings given by others, or anxiety due to the expectation of negative evaluations from the side. Also, probably, there is an incompatibility of different systems of requirements from parents and the school, shown to the child.

A significant reduction in scales (1, 3, and 7) confirms the effectiveness of therapeutic efforts in the personal characteristics of the child and improvement of situations associated with schooling.

The next observed child’s anxiety was the anxiety associated with schooling (scales 6 and 8). The average anxiety index of the median on scale 6 in group II after remedial measures significantly decreased by 2 points (*p* < 0.05), which indicates a decrease in fear of something new, an increase in motivation to achieve success (Table 1). On scale 8, the group average anxiety of the median arising in situations of knowledge testing, was seen to decrease by 1 point and the interquartile range also became significantly lower (C_25_; C_75_). This testifies that the fear of a public demonstration of their knowledge proved to decrease, and the level of student confidence increased.

The reduction of anxiety in group II by five scales indicates that, after the SPT and the BFB training, the general emotional state of the students stabilized, and self-esteem increased. The children were becoming more confident in themselves and their abilities. The sensitivity to the estimates of others proved to decrease. In presenting themselves, their abilities, the students became bolder and more confident.

Analyzing the impact of therapeutic efforts on the parameters of child anxiety in group III (Table 1), significant changes are visible in most of the scales of MACA. The average group median decreased by 1.5–3.5 points (*p* < 0.05–*p* < 0.01) within 7 out of 10 scales (scales 1, 2, 5, 6, 7, 9, 10).

Scales 2, 4, and 5 reflect anxiety in a child’s social contacts with peers, teachers, and parents. On scale 2 (anxiety in relationships with age mates), there was a significant decrease in the indicator (*p* < 0.01). This fact indicates a decrease in anxiety and fear about criticism and rejection by peers, as well as stabilization of the emotional state of the child. On scale 4 (anxiety in relationships with teachers), significant changes were not identified. No therapeutic efforts in this direction were conducted, although the literary data indicate a connection between the consistently elevated levels of anxiety of younger schoolchildren with the teacher, individual, and personal characteristics [21]. On scale 5 (anxiety in relationships with parents), the median decreased by 2.5 points (*p* < 0.01). In our study, no work with parents was carried out either (although it was offered). Apparently, a decrease in general anxiety, the development of self-regulation skills indirectly influenced the child–parent relationship and the emotional situation at home significantly improved.

## 4. Discussion

It is important to note that the performed therapeutic efforts positively affected the indicators on scales 9 and 10, which reflect the characteristics of the psychophysiological and psychoautonomic anxious response of the child in stressful situations. It is known that overwork as a result of school load leads to failures, and the accumulated experience of failures causes fear, and fuels the anxiety. Internal conflict is resolved at the cost of health loss [19]. The probability of a psychosomatic response to an alarming factor in the environment is very high. This can result in frequent colds, gastrointestinal disorders and cardiovascular dysfunction, headaches, odontophobia and allergic reactions after or before stressful situations [19,22,23]. The results of the analysis of the MACA questionnaire indicate that, after the conducted therapeutic efforts, schoolchildren of group III increased their adaptability to stressful situations, decreased fatigue, irritability, improved sleep and appetite, and the body’s performance in general (*p* < 0.05–*p* < 0.01).

Note that it is the anxiety arising in situations of self-expression (scale 7) that closely correlates with all scales of MACA (r = 0.76–0.93; *p* < 0.01). It is the fear of self-expression that is most closely associated with the rest of the structures of school anxiety. The need to demonstrate their capabilities and self-doubt are the basis of school anxiety. In this regard, the student intentionally seeks to avoid cases related to the experience of new relationships, prefers to work in a group, and often is passive and dependent [19].

When analyzing the therapeutic efforts which we carried out, it is clear that anxiety reduction in each group occurred by different scales (Table 1), and, the more remedial measures varied, the more changes were noted. However, regardless of the therapy methods, anxiety in scale 7 that was the anxiety associated with the fear of self-expression proved to decrease throughout all the studied groups. Thus, all the therapy methods affected the personality characteristics of the child associated with the need for self-disclosure, presenting oneself to others, and demonstrating one’s capabilities. Regardless of the set of therapeutic efforts performed, the participants of the three groups became more resolute, independent, active, and took more initiative. Considering that scale 7 correlates better than others with the rest of the structures of school anxiety, it can be assumed that by conducting targeted work to reduce anxiety that arises in situations of self-expression, anxiety will be corrected in other areas.

Also, in our study, the effect of all the therapeutic efforts on the anxiety in scale 6 can be clearly visible, which value decreased by 1.5–2.0 points in all the groups (*p* < 0.05–*p* < 0.01). That is, the psychological background that allows for developing the needs in achieving high results became more favorable for the children. Anxiety associated with academic performance and in the performance of training tasks was observed to decrease.

Our data are consistent with studies [21] which indicate that children of primary school age have a high level of general school anxiety due to fear in a situation of knowledge testing, fear of self-expression, and low physiological resistance to stress.

## 5. Conclusions

As a result, our study showed that therapeutic efforts such as sociopsychological training, the biofeedback method, and Sand Therapy are effective ways that reduce anxiety and, in general, improve the psychoemotional state of younger schoolchildren.

It was revealed that the group which underwent various remedial measures demonstrated the greatest number of changes: anxiety significantly decreased according to the 7 scales of the children’s anxiety questionnaire: general anxiety; anxiety in peer relations; anxiety in relationships with parents; anxiety associated with success in learning; anxiety arising in situations of self-expression; decreased mental activity associated with anxiety; increased autonomic reactivity associated with anxiety. The combination of various methods of therapy gives the best results in reducing anxiety.

However, it is important to remember that the effectiveness of therapy programs essentially depends on the active participation of the actual student in the classes. Also, the success of the resolution of school anxiety affects the early detection of irregularities in the development of the student and the timely provision of remedial assistance. The earlier problems in the adaptation of the younger student are identified, the less time will be required for its correction.

In order to preserve and improve the mental and physical condition of schoolchildren, it is necessary to introduce psychoprophylactic measures in the educational process and conduct health-improving lessons using group and individual therapeutic efforts: sociopsychological training, biofeedback and art-therapeutic techniques. The complex application of therapeutic efforts is quite promising in reducing the child’s anxiety.

## Figures and Tables

**Table 1 behavsci-10-00020-t001:** Anxiety indices in younger schoolchildren before and after the therapy sessions, (Me, 25; 75 percentile).

MACA Scales, Point	Group I	*p*	Group II	*p*	Group III	*p*
	n = 12		n = 11		n = 20	
	Me (C_25_; C_75_)		Me (C_25_; C_75_)		Me (C_25_; C_75_)	
1	5.0 (2.0; 6.0)	T = 8	5.0 (3.0; 7.0)	T = 13	4.5 (2.8; 7.3.)	T = 14
	3.5 (0.8; 5.3)		1.0 (0.5; 6.0) *	*p* < 0.05	3.0 (1.0; 5.0) *	*p* < 0.05
2	4.0 (2.8; 5.3)	T = 10.5	3.0 (2.0; 6.0)	T = 21	4.5 (2.0; 6.0)	T = 31
	2.5 (1.8; 4.0) *	*p* < 0.05	3.0 (1.5; 4.5)		2.5 (1.8; 3.3) *	*p* < 0.05
3	5.5 (4.3; 7.0)	T = 21	6.0 (2.5; 7.0)	T = 7.5	4.0 (3.0; 6.0)	T = 49
	3.5 (3.0; 5.3)		4.0 (1.5; 4.0) *	*p* < 0.05	3.5 (2.0; 5.0)	
4	3.5 (2.0; 4.3)	T = 26.5	4.0 (2.5; 5.5)	T = 21.5	4.5 (3.0; 6.3)	T = 60
	4.0 (2.0; 5.0)		4.0 (2.0; 5.5)		4.0 (1.0; 6.0)	
5	4.5 (3.8; 6.0)	T = 50	4.0 (3.0; 5.0)	T = 25	3.5 (3.0; 5.3)	T = 29
	4.0 (3.0; 4.3)		4.0 (2.0; 5.0)		2.0 (2.0; 3.3) **	*p* < 0.1
6	5.0 (4.0; 6.3)	T = 8.5	5.0 (3.0; 6.0)	T = 2	5.0 (4.0; 6.3)	T =17
	3.5 (2.0; 4.3) *	*p* < 0.05	3.0 (2.0; 4.0) *	*p* < 0.05	3.5 (1.0; 5.0) **	*p* < 0.01
7	5.0 (4,8; 6.3)	T = 1	4.0 (2.5; 5.0)	T = 10	6.5 (2.8; 7.0)	T = 15.5
	3.0 (2.8; 5.0) **	*p* < 0.01	3.0 (1.5; 3.0) *	*p* < 0.05	3.0 (1.8; 4.3) **	*p* < 0.01
8	3.5 (3.0; 6.3)	T = 30.5	5.0 (3.5; 6.0)	T = 1.5	5.0 (4.0; 6.0)	T = 53.5
	3.5 (1.8; 7,3)		4.0 (2.0; 4.5) *	*p* < 0.05	5.0 (3.0; 6.0)	
9	3.0 (2.8; 4.3)	T = 19.5	4.0 (2.5; 5.5)	T = 22.5	5.0 (3.8; 6.0)	T = 26
	2.5 (1.0; 5.0)		4.0 (3.0; 4.5)		2.5 (1.0; 5.0) **	*p* < 0.01
10	3.5 (1.8; 5.3)	T = 16	5.0 (3.0; 6.5)	T = 12.5	5.0 (2.8; 6.0)	T = 24.5
	2.0 (1.0; 6.0)		2.0 (0.5; 3.5)		2.5 (0.0; 5.0) *	*p* < 0.05
IIA	39.0 (34.0; 55.3)	T = 13.5	49.0 (35.5; 52.5)	T = 12	46.0 (33.8; 57.0)	T = 22
	29.0 (20.3; 49.0) *	*p* < 0.05	28.0 (22.0; 46.0) *	*p* < 0.05	32.0 (20.0; 42.5) **	*p* < 0.01

MACA (Method of Multidimensional Assessment of Child Anxiety) scales: 1 is for general anxiety; 2 is anxiety in relationships with age mates; 3 is anxiety in connection with the assessment of others; 4 is for anxiety in relationships with teachers; 5 is for anxiety in relationships with parents; 6 is anxiety associated with success in studying; 7 is for anxiety arising in situations of self-expression; 8 is anxiety arising in situations of testing knowledge; 9 is a decrease in mental activity associated with anxiety; 10 is for anxiety increase in autonomic reactivity. Group I is students underwent only social psychological training. Group II is students underwent the SPT (sociopsychological training) and “BOS-Pulse” sessions (biofeedback method (BFB). Group III is students involved in SPT, “BOS-Pulse”, and Sand Therapy sessions. Above the line is data before the therapy sessions, under the line is data after the therapy sessions. * is for significant changes after the therapeutic efforts within the group at *p* < 0.05; ** is for significant changes after the therapeutic efforts within the group at *p* < 0.01.

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
