# Peer review of "The Effectiveness of Various Types of Psychological Correction of Anxiety in Primary School"

_behavsci, 2019, doi:10.3390/bs10010020_

Round 1

Reviewer 1 Report

As the authors point out, the reduction of anxiety in children is important because anxiety has significant influence on the development of later mental and physical disorders.  Therefore, the study is of importance and I believe will be of interest to readers.  However, the manuscript needs to be edited by a professional proofreader who can improve the English. It is difficult to grasp right now. 

Two considerations:

“corrective measures” is a term used in English more in the in the juvenile justice system than psychologically. The term “psychocorrection” is not used in English.  I would use “therapeutic efforts” or “psychoeducation” instead if your target audience is in the US and Europe.

If you use an abbreviation the abbreviation Me for median, then do not go back to using the word. However, I would use the word and not the abbreviation because Me is not common in English.

In addition, it would be of interest to the readers if the content of the sessions was more clearly explained.  As it stands with the relatively vague descriptions of session content, no one could replicate your study.

on page 3, lines 143-146 are not necessary:

Literature searches (PubMed and Google Scholar) were carried out (2000 to 2018) using the keyword «psychocorrection» and cross-referencing it with «physicians», «depression», «younger school children», «socio-psychological training», «biological feedback» and «art-therapy».

Author Response

“corrective measures” is a term used in English more in the in the juvenile justice system than psychologically. The term “psychocorrection” is not used in English.  I would use “therapeutic efforts” or “psychoeducation” instead if your target audience is in the US and Europe.

Response. The text was changed“ "corrective measures" was replaced by "therapeutic efforts".

If you use an abbreviation the abbreviation Me for median, then do not go back to using the word. However, I would use the word and not the abbreviation because Me is not common in English.

Response. Replaced the abbreviation Me with the word "median".

In addition, it would be of interest to the readers if the content of the sessions was more clearly explained.  As it stands with the relatively vague descriptions of session content, no one could replicate your study.

Response. Added a more extensive description of the method of biological feedback in the section Materials and Methods, lines 130-134

“In order to win the competition, the player must reduce the heart rate by learning to manage their own self-regulation mechanisms in combination with a high degree of control of consciousness. Along with the method of biological feedback the Progressive muscle relaxation according to Jacobson was applied using breathing techniques and images.”

on page 3, lines 143-146 are not necessary:

Literature searches (PubMed and Google Scholar) were carried out (2000 to 2018) using the keyword «psychocorrection» and cross-referencing it with «physicians», «depression», «younger school children», «socio-psychological training», «biological feedback» and «art-therapy».

Response. Removed.

Reviewer 2 Report

Dear Authors, this manuscript is really interesting and it is about a current topic.

The manuscript is well conducted but it needs improvements.

In abstract section please enlarge "intro and background" before the aim.

In keyword section please use medical subject heading MESH word.

The manuscript could take advantage on readability by subdviding introduction section into 2 subsection (intro and aim).

Please subdivide M&M section too for a better readability (patient selection, incl and excl criteria, method, outcome, statistical analysis etc)

In discussion section please cite a study about fear and anxiety on child patients:

De Stefano, R.; Bruno, A.; Muscatello, M.; Cedro, C.; Cervino, G.; Fiorillo, L. Fear and anxiety managing methods during dental treatments: systematic review of recent data. Minerva Stomatol 2020.

De Stefano, R. Psychological Factors in Dental Patient Care: Odontophobia. Medicina 2019, 55, 678.

In conclusion section please provide more future perspective of this study.

Thank you 

Author Response

In abstract section please enlarge "intro and background" before the aim.

Response In the abstract added enlarge "intro and background". Lines 10-15.

“At various stages of development and education of children, there are psycho-emotional difficulties that create prerequisites for impairment of the development of the child’s personality. The timely detection and therapeutic efforts of difficulties in schoolchildren is important for the formation of a psychologically healthy personality. The study of the effectiveness of various remedial techniques for childhood anxiety has therefore become theoretically and practically significant”.

In keyword section please use medical subject heading MESH word.

Response In keyword and text replaced "corrective measures" with "therapeutic efforts".

The manuscript could take advantage on readability by subdviding introduction section into 2 subsection (intro and aim).

Please subdivide M&M section too for a better readability (patient selection, incl and excl criteria, method, outcome, statistical analysis etc)

Response Added the sections.

Materials and Methods

2.1. Research design

2.2. Instruments

2.3. Statistical analysis

In discussion section please cite a study about fear and anxiety on child patients:

De Stefano, R.; Bruno, A.; Muscatello, M.; Cedro, C.; Cervino, G.; Fiorillo, L. Fear and anxiety managing methods during dental treatments: systematic review of recent data. Minerva Stomatol 2020.

De Stefano, R. Psychological Factors in Dental Patient Care: Odontophobia. Medicina 201955, 678.

Response In discussion section added to the article: De Stefano, R. Psychological Factors in Dental Patient Care: Odontophobia. Medicine.  2019, 55, 678. 

Lines 225-227. “This can result in frequent colds, gastrointestinal disorders and cardiovascular dysfunction, headaches, odontophobia and allergic reactions after or before stressful situations [19, 22].”

And also included in the References paragraph No. 22. Lines 329.

In conclusion section please provide more future perspective of this study.

Response Added a phrase. Lines 274-275. The complex application of therapeutic efforts is quite promising in reducing the child anxiety.

Round 2

Reviewer 2 Report

Dear Authors, Your manuscript has been improved now.

I'm asking again if You could add the first proposed reference (Min Stomatol; in press article).

thank You

kind regards

Author Response

Response to Reviewer 2 Comments

 I'm asking again if You could add the first proposed reference (Min Stomatol; in press article).

Response.  In discussion section added to the article: De Stefano, R.; Bruno, A.; Muscatello, M.; Cedro, C.; Cervino, G.; Fiorillo, L. Fear and anxiety managing methods during dental treatments: systematic rerview of recent data. Minerva Stomatol 2020.

Lines 229 “This can result in frequent colds, gastrointestinal disorders and cardiovascular dysfunction, headaches, odontophobia and allergic reactions after or before stressful situations [19, 22, 23].”

And also included in the References paragraph No. 23. Lines 333-334.
